# Asthma and Chronic Rhinosinusitis: Diagnosis and Medical Management

**DOI:** 10.3390/medsci7040053

**Published:** 2019-03-27

**Authors:** Landon Massoth, Cody Anderson, Kibwei A. McKinney

**Affiliations:** College of Medicine, University of Oklahoma, Oklahoma City, OK 73104, USA; landon-massoth@ouhsc.edu (L.M.); cody-anderson@ouhsc.edu (C.A.)

**Keywords:** asthma, sinusitis, chronic rhinosinusitis, nasal polyps, eosinophilia, sinus surgery

## Abstract

Asthma is a prevalent inflammatory condition of the lower airways characterized by variable and recurring symptoms, reversible airflow obstruction, and bronchial hyperresponsiveness (BHR). Symptomatically, these patients may demonstrate wheezing, breathlessness, chest tightness, and coughing. This disease is a substantial burden to a growing population worldwide that currently exceeds 300 million individuals. This is a condition that is frequently encountered, but often overlooked in the field of otolaryngology. In asthma, comorbid conditions are routinely present and contribute to respiratory symptoms, decreased quality of life, and poorer asthma control. It is associated with otolaryngic diseases of the upper airways including allergic rhinitis (AR) and chronic rhinosinusitis (CRS). These conditions have been linked epidemiologically and pathophysiologically. Presently, they are considered in the context of the unified airway theory, which describes the upper and lower airways as a single functional unit. Thus, it is important for otolaryngologists to understand asthma and its complex relationships to comorbid diseases, in order to provide comprehensive care to these patients. In this article, we review key elements necessary for understanding the evaluation and management of asthma and its interrelatedness to CRS.

## 1. Introduction

Asthma is a prevalent inflammatory condition of the lower airways characterized by variable and recurring symptoms, reversible airflow obstruction, and bronchial hyperresponsiveness (BHR). Symptomatically, these patients may demonstrate wheezing, breathlessness, chest tightness, and coughing. This disease is a substantial burden to a growing population worldwide that currently exceeds 300 million individuals [1]. This is a condition that is frequently encountered, but often overlooked in the field of otolaryngology. In asthma, comorbid conditions are routinely present and contribute to respiratory symptoms, decreased quality of life, and poorer asthma control. It is associated with otolaryngic diseases of the upper airways including allergic rhinitis (AR) and chronic rhinosinusitis (CRS) [2,3]. These conditions have been linked epidemiologically and pathophysiologically. Presently, they are considered in the context of the unified airway theory, which describes the upper and lower airways as a single functional unit [4,5]. Moreover, the upper and lower respiratory tracts share anatomical and histological characteristics. They have common histological structures, including the basement membrane, lamina propria, ciliary epithelium, glands, and goblet cells [6]. Because of these similarities, it is important for otolaryngologists to understand the complex relationships that exist between asthma and comorbid diseases, in order to provide comprehensive care to these patients. In this article, we review key elements necessary for understanding the evaluation and management of asthma and its interrelatedness to CRS.

## 2. Epidemiology

Asthma is a common condition affecting greater than 4% of the population globally [7]. It is more prevalent in developed countries, and, in particular, the United States [8]. Over the last several decades, there has been a rise in its prevalence. Epidemiologic studies by the Centers for Disease Control (CDC) reported a 3.0% asthma prevalence in the United States in 1970, which rose to 7.8% by 2006 to 2008 [9]. Today, it affects 8.4% of children and 8.1% of adults in the United States [10,11]. It is a leading cause for presentation to the emergency department, accounting for 1.7 million visits in the United States in 2015 [12]. The rate of asthma deaths decreased from 15 per million in 2001 to 10 per million in 2016. Adults were nearly five times more likely than children to die from asthma. There remain gender and racial disparities in morbidity and mortality among different groups. This is demonstrated by the higher death rate among women and non-Hispanic blacks, with the latter group being two to three times more likely to die from asthma when compared with other racial groups [13].

From a pathophysiology standpoint, the current understanding is that certain gene–environment and gene–gene interactions contribute to the development of asthma. Tobacco smoke exposure, pollutants, respiratory viral infections, and obesity are significant risk factors in its pathogenesis [14,15,16,17]. Hereditary factors play an apparent role in the development of the disease as well. In a Swedish study, a family history of atopic asthma increased the risk of developing this condition up to four-fold [18]. Moreover, several studies have shown that the offspring of asthmatic parents are at an increased risk of developing asthma [19]. The list of genes associated with asthma continues to grow, as more are elucidated through whole genome sequencing.

Epidemiologic evidence also supports the coexistence of asthma and other upper airway conditions, as previously mentioned. For instance, nearly 80% of patients with asthma report some form of rhinitis, defined as irritation and inflammation of the mucous membranes of the nose. Conversely 10–40% of rhinitis patients report coexistent asthma [20,21]. The presence of rhinitis increases the risk for the development of asthma by three-fold in both atopic and non-atopic individuals [22]. In a study by Linneberg et al., individuals with AR who were sensitized to perennial allergens were found to have a significantly higher likelihood of developing asthma than individuals who were sensitized to seasonal allergens [23].

The prevalence of CRS is estimated to be between 22% and 45% among patients with asthma [24,25,26]. Among the general population, the prevalence of CRS symptoms is estimated to be 10–12%, with the majority of these patients reporting either moderate or severe symptoms [27]. CRS is associated with more severe asthma, especially in patients with nasal polyps (CRSwNP) [28]. The presence of nasal polyps is similarly associated with more severe sinus symptoms, including facial pain and pressure and hyposmia, relative to CRS without polyps (CRSsNP) [27]. In a recent cluster analysis, performed by the Severe Asthma Research Program (SARP), nearly half of the patients with the most severe burden of disease had a history of previous sinus surgery [26].

## 3. Pathophysiology of Asthma

The hallmark characteristics of asthma pathogenesis include inflammation of the lower airways through the infiltration of cells, release of potent pro-inflammatory factors, and the remodeling of the airway walls. Allergens, pollutants, irritants, and microbes elicit various inflammatory cascades that are mediated by multiple cell types including dendritic cells, mast cells, eosinophils, T lymphocytes, macrophages, neutrophils, and epithelial cells [29]. These inflammatory influences provoke changes in the respiratory tract, including epithelial shedding, goblet cell hyperplasia, hypertrophy of submucosal mucus glands and bronchial smooth muscle, subepithelial fibrosis with collagen deposition, angiogenesis, and vascular permeability [29]. This is manifested by variable and recurrent episodes of wheezing, breathlessness, chest tightness, and cough. Initially, these changes are reversible with treatment, but with chronic inflammatory insults, irreversible remodeling of the lower airway occurs. These alterations increase the thickness of the airway wall, leading to irreversible airflow obstruction and airway hyperresponsiveness. Similar histopathological changes are often observed in CRS, including mucosal thickening, submucosal gland hypertrophy, collagen deposition, and basement membrane thickening [30].

While inflammation is central to the pathophysiology of asthma, the underlying mechanism is increasingly understood to be multifactorial, reflecting the diversity of the natural history, severity, and treatment responsiveness of the disease. Presently, asthma is seen as an umbrella diagnosis with several distinct mechanistic pathways (endotypes) and variable clinical presentations (phenotypes). While there is little consensus, endotypes in asthma are generally categorized as T-helper type 2 (Th2) cell—high (T2–high) or Th type 2 cell—low (T2–low) [31]. The former is characterized by eosinophilia and atopy, while the latter is manifested by increased neutrophils or a pauci-granulocytic profile. T2–high inflammation is associated with the eosinophilic airway reactivity that is driven by dendritic cell stimulation of Th2 cells and the production of inflammatory cytokines [32]. Allergen exposure leads to the production of Interleukin-33 (IL-33), IL-25, and thymic stromal lymphopoietin (TSLP) by both dendritic and epithelial cells. These mediators, recently classified as alarmins, stimulate Th2 cells to release IL-4, IL-5, and IL-13 that, in turn, stimulate eosinophils, mast cells, and immunoglobulin E (IgE) synthesis through the induction of IgE B cells. Immune memory of IgE responses is then maintained by the development of plasma cells contained in the bone marrow. Recent studies suggest that the respiratory mucosa is the site of development of these cells that maintain immunologic memory of allergen-induced IgE responses [33]. The authors would direct the reader to consider contemporary reviews regarding the biologic development of IgE and immunologic memory in allergic airway disease [33,34].

Nonallergic irritants produce an analogous inflammatory cascade by stimulating the production of IL-33, IL-25, and TSLP, which lead type 2 innate lymphoid cells to produce IL-5 and IL-13. Unlike Th2 cells, type 2 innate lymphoid cells produce little IL-4 and do not elicit an IgE response. Alternatively, in T2–low inflammation, evidence suggests that irritants, pollutants, or infectious agents activate IL-33, IL-23, and IL-6 [35]. IL-33 stimulates the Th17 cell to produce IL-6, IL-17, and IL-8. These cytokines, in turn, trigger neutrophil production. Th17 and Th1 activate neutrophilic inflammation through IL-6, IL-17, interferon gamma, and tumor necrosis factor alpha [35]. A growing understanding of these mechanisms has coincided with the advent of treatment strategies that target specific inflammatory mediators, based on biomarkers that reflect the underlying disease. It is the hope of investigators that further insight will lead to tailored therapy and improved outcomes.

## 4. Pathophysiology of CRS

The pathogenesis of CRS parallels that of asthma. Specifically, the inflammatory subtypes of CRS mirror T2–high and T2–low inflammatory endotypes observed in asthma [36]. Historically, CRS was categorized as either CRS with nasal polyps or CRS without nasal polyps (CRSsNP), based on the presence or absence of polyps on imaging or sinonasal endoscopy [37]. CRSwNP is generally accepted to have a type 2-predominant inflammatory response, with a predominance of eosinophilic inflammation, including eosinophils, mast cells, elevated IgE, and the expression of type 2 cytokines (IL-4, IL-5, IL-9, IL-13, IL-25, and IL-33). Consequently, CRSwNP has a close association with asthma and other atopic diseases. CRSsNP, on the other hand, is more commonly associated with elevated type 1 cytokines (e.g., interferon-γ), Th-1 helper cells, and a neutrophilic cellular response [37,38]. Recent studies have demonstrated immense heterogeneity between and within these broad phenotypic categories, based on the molecular and cellular pathways that are active in each specific disease state [37,38,39]. These “endotypes” differ in the severity of disease and demonstrate histopathologic differences in various inflammatory cascades that are the potential targets of therapeutic intervention.

Cluster analyses of the factors contributing to CRS have been performed, taking into consideration the clinical, molecular, and pathological markers of disease [37,40,41,42]. This has been particularly helpful in differentiating the underlying mechanisms of disease in patients with nasal polyps. Cao et al., for example, found that type 2 immune pathway activation was a predisposing factor toward the development of nasal polyps in both patients with and without eosinophilic inflammation [40]. In another study, Tomassen et al. clustered patients into groups according to their levels of IL-5 expression, as a proxy for the level of eosinophilic inflammation present [43]. In doing so, low levels of IL-5 were shown to have a close association with CRSsNP, while high levels correlated with the highest burden of polyps. Intermediate levels of IL-5 demonstrated variable phenotypic expression of asthma and nasal polyposis, suggesting that there is a subset of CRSwNP patients that exhibits both type 1 and type 2 immune pathway activation [37,38,43]. Nakayama et al. similarly identified disease-specific factors that correlated with clusters of disease, including the presence of perennial allergy, asthma/eosinophilic mucin, and eosinophilic inflammation [44]. In a much larger study of over 100 patients, Soler et al. determined that other patient-related factors were more discriminant in clustering patients with CRSwNP, including age, patient-reported outcome measures (Sinonasal Outcome Test-22 (SNOT-22) scores, productivity loss), and the comorbid presence of fibromyalgia and depression [45]. Ongoing investigation into the relative contributions of patient- and disease-related factors is underway; with the goal of determining how these factors interact to influence the severity of disease and the response to directed therapies.

## 5. Clinical Features of Asthma

Classically, the cardinal symptoms of asthma include wheezing, breathlessness, chest tightness, and coughing. The clinical features and severity of asthma, however, can vary significantly between individuals and even in a single person over time [46]. Additionally, many asthmatics may only recognize fulminant asthma exacerbation as an indicator of this disease, with a tendency to overlook more subtle, indolent symptoms (e.g., nighttime cough), leading to delays in diagnosis and treatment [47]. Cough associated with asthma classically worsens at night or with activity, is dry, and non-productive. In one Korean study, 680 adult patients were questioned on whether they experience a troublesome cough at night. The association of this symptom with the diagnosis of asthma demonstrated a sensitivity of 62.1%, specificity of 44.8%, positive predictive value (PPV) of 22.8%, and negative predictive value (NPV) of 81.8% [48]. Wheezing is also a common clinical feature of asthma. A large New Zealand study showed that wheezing had a sensitivity of 94% and specificity of 76% for the diagnosis of asthma. Furthermore, this study demonstrated that wheezing with dyspnea was the best predictor of asthma with a sensitivity and specificity of 82% and 90%, respectively [49]. The New Zealand study also examined exercise dyspnea alone. Exercise-induced dyspnea was found to have a sensitivity of 75.4%, specificity of 76.5%, PPV of 22.5%, and NPV of 97.2% [50]. It is, therefore, critical for the otolaryngologist to both inquire about these symptoms and consider further evaluation for asthma when the symptoms present.

Due to the heterogeneity of the disease, a number of asthma phenotypes have been proposed. Phenotypes have been organized according to their association with specific triggers, patient characteristics, or features of clinical presentation. Examples include aspirin sensitivity, adult age of onset, and steroid-resistant subtypes. This heterogeneity prompted the National Heart, Lung, and Blood Institute’s Severe Asthma Research Program (SARP) to perform a cluster analysis on adults with mild, moderate, and severe asthma to identify phenotypic clusters which share common traits. The analysis revealed five phenotypic clusters that were predominantly distinguished by lung function and disease age of onset [26]. A similar cluster analysis by SARP identified four phenotypic clusters in children that differed primarily according to asthma duration, the number of asthma controller medications being used, and lung function [51]. Of these clusters, late-onset asthmatics had higher frequency of sinusitis, more severe sinus disease radiographically, and higher rates of sinus surgery [52,53,54]. Accordingly, it is crucial in these patients to assess for cardinal symptoms associated with CRS, including nasal obstruction, discolored nasal discharge, facial pain or pressure, loss of sense of smell, and cough (in children) that persists for greater than 12 weeks [55,56]. There is growing momentum to pair these clinical phenotypes with pathophysiologic mechanisms. To date, however, the association between clinical phenotype and endotype remains imprecise and, thus, warrants further investigation.

## 6. Clinical Features of Chronic Rhinosinusitis

Recent studies have demonstrated the immense heterogeneity represented among patients previously categorized under the broad diagnosis CRS with nasal polyps. This has derived from the realization that patients with this phenotype differ in the extent to which tissue and blood eosiniophilia play a role in their pathogenesis. Similar to eosinophilic asthma, eosinophilic chronic rhinosinusitis (eCRS) is associated with aberrant Th2 pathway activation, resulting in activation of the eosinophilic inflammatory cascade and mucociliary dysfunction. These patients tend to manifest a more severe disease presentation, with symptoms that are more refractory to medical and surgical interventions than non-eCRS subtypes. These symptoms include nasal obstruction, olfactory dysfunction, thick mucus drainage, and recurrent episodes of bacterial infection [57].

In a recent review of the literature, Dennis et al. identified four specific classification schema that have been used to endotype patients with CRSwNP. In the type 2 cytokine-based approach, endotypes are differentiated based upon the activation of the Th1 or Th2 pathway. The eosinophil-based approach looks to characterize the endotype of CRSwNP based on the presence of an eosinophilic versus neutrophilic infiltrate within the sinonasal mucosa. This approach accounts for the fact that although these patients manifest a similar phenotype, the degree of eosinophilic inflammation may vary widely between disease processes and among different populations. Eosinophilc mucin, for example, is closely associated with AERD, allergic fungal rhinosinusitis, and eosinophilic mucin chronic rhinosinusitis (EMCRS), but is less often affiliated with non-eosinophilic CRSwNP. A third strategy has looked at the levels of IgE as a marker for different endotypes of CRSwNP. Elevated IgE is noted in all endotypes of patients with polyp disease, with the exception of AERD. However, it is now recognized that local rather than systemic IgE may play a greater role in the development of tissue eosinophilia. More specifically, the production of *Staphylococcus aureus* enterotoxin-specific IgE has been found to correlate more closely with local IgE concentrations and asthma [37,39,43].

Finally, the Cysteinyl Leukotriene (CysLT)-based approach acknowledges aspirin exacerbated respiratory disease (AERD) as a unique clinical phenotype of CRSwNP that is associated with asthma and intolerance of cyclogoxygenase-1 inhibiting agents [37,43,58]. This disease is considered to be a hypersensitivity reaction to acetylsalicylic acid and cyclooxygenase (COX)-1-inhibiting non-steroidal anti-inflammatory drugs that first manifests with nasal congestion and rhinorrhea, typically during the second decade of life. Over several years, it evolves into a more severe and recalcitrant form of disease, that eventually progresses to affect both the upper and lower airways in the form of CRS with nasal polyposis and asthma [59]. The elevated levels of CysLT are due to a functional deficit of COX enzymes and hyperactivity of the 5-lipoxycgenase and leukotriene C4 synthase pathways, resulting in overexpression CystLT [58,59]. A meta-analysis from 2015 found the prevalence of AERD to be 7% in patients with classical asthma and 14% in patients with severe asthma [60]. It also accounts for almost 10% of all patients with CRSwNP [37]. The presence of nasal polyps in a patient with severe asthma should, therefore, prompt the otolaryngologist to consider this particular variant of asthma in their treatment approach.

## 7. Asthma Diagnosis and Assessment

Asthma can be difficult to diagnose due to its high clinical variability and episodic nature. The diagnosis of asthma is best accomplished through a comprehensive history and physical examination, combined with objective pulmonary function testing [61]. In addition to inquiring about the cardinal symptoms, it is equally important to assess the patient for other risk factors such as smoking, tobacco exposure, family history, and other signs of atopy. Children of parents who are both affected by asthma have an 6.7-fold increased relative risk of asthma when compared to children without any family history [50]. Relying on the physical examination for the diagnosis of asthma also has its challenges. Patients will often present with normal vital signs and physical findings [46]. Moreover, respiratory physical exam findings can be examiner-dependent, as studies have shown only fair-to-good inter-examiner reliability in detecting wheezing on auscultation [62].

Objective pulmonary function testing is considered the gold standard for the definitive diagnosis of asthma. Two findings need to be present with objective diagnostic testing for asthma: (1) the presence of airway obstruction, demonstrated by a decreased forced expiratory volume in one second (FEV1) to forced vital capacity (FVC) ratio, and (2) variability in the severity of airway obstruction when subjected to bronchodilatory or bronchoconstrictive stimuli [61,63]. Spirometry is the objective pulmonary testing method of choice. Using spirometry, an obstructive airway pattern can be established when FEV1/FVC is less than 0.75 in adults or 0.9 in children. Excessive variability in lung function is demonstrated by an increase or decrease of FEV1 greater than 12% after a bronchodilator reversibility test or four week trial of anti-inflammatory treatment [63].

Other supportive testing methods may also be employed. For example, bronchial provocation using exercise or methacholine with measurement of the fractional concentration of exhaled nitric oxide (FeNO) may be employed if the initial spirometry tests are negative and clinical suspicion remains high [46,63]. Additionally, diagnosis of allergic asthma may rely on allergy testing such as skin testing and in vitro ImmunoCAP IgE tests to exclude or confirm the presence of atopy [64]. Finally, a burgeoning area of diagnostic testing is the use of predictive biomarkers in the diagnosis of asthma. Currently, common biomarkers include aberrations in FeNO, serum IgE, sputum and blood eosinophil count, and serum periostin. In particular, FeNO has become more widely available and is a non-invasive reflection of airway eosinophilia. It is useful as a marker of adherence to therapy as well as a predicator of upcoming exacerbation [65]. Recent systematic reviews also demonstrated that tailoring therapy based on FeNO levels may also reduce the number of asthma exacerbations in adults and children [66,67]. Conceptually, the use of biomarkers has the potential advantage of enabling the identification of specific clinical phenotypes and individualizing therapy, with the aim of improving patient outcomes [68].

## 8. Chronic Rhinosinusitis Diagnosis and Assessment

Chronic sinusitis has been defined as the presence of ≥2 of the following symptoms for ≥12 weeks duration: anterior or posterior nasal drainage, nasal obstruction, hyposmia or anosmia, and/or facial pain and pressure. These symptoms must be correlated with objective evidence of mucosal disease, including endoscopic evidence of purulence, edema, or nasal polyposis, and/or mucous membrane thickening on computed tomography imaging [55,58].

Although inflammatory markers are helpful, the diagnosis of CRS is still largely contingent upon self-reported symptoms and computed tomography (CT) findings demonstrating polyp disease, mucous membrane thickening, and ostiomeatal obstruction, with the posterior ethmoid and olfactory cleft being the anatomic regions most predictive of this disease process [69]. The Lund–Mackay scoring system has long been used as an objective measure of disease severity, with each sinus being graded on a scale of 0 to 2, based on the degree of mucosal thickening present on CT imaging. The sinuses are grouped into six anatomic regions on each side of the nose, giving a total possible score of 24. These scores provide useful information about the location and extent of diseased tissues, but poorly predict patient symptoms [70]. Self-reported symptoms include the use of validated quality of life surveys, including the SNOT-22 and nasal symptom score (NSS) and rhinosinusitis disability index (RSDI), among many others. While effective as a screening measure, the accuracy of self-reported symptoms is low with a sensitivity and specificity for predicting CRS of 84% and 82%, respectively [71].

Based on the expanding knowledge of CRS endotypes, more research is being done to evaluate the utility of these markers in the diagnosis, work-up, and long-term management of these patients. In addition to the use of IL-5 (which has garnered a great deal of attention due to the proliferation of anti-IL-5 commercially-available monoclonal antibodies), other markers being examined include epithelial-derived cytokines (IL-25 and IL-33), type 2 innate lymphoid cells (ILC2s), cytokines that promote type 2 adaptive responses (IL-4 and IL-13), and the measurement of urinary Leukotriene C4 (LTC_4_,) in the case of suspected AERD [37,38,39,40,43]. Other studies have advocated for the use of structured histopathologic analysis of excised sinonasal tissues, to determine the type of inflammatory infiltrate and the extent of tissue remodeling that may indicate disease severity and predict response to treatment [39]. Among these are basement membrane thickening, subepithelial edema, mucosal ulceration, and fibrosis. To date, none of these histopathologic features have reliably differentiated between the different endotypes of CRS [39].

Similar consideration has been given to utilizing serologic and tissue biomarkers as predictors of disease more globally among all eCRS pathways. Although no strict diagnostic criteria of eCRS exists, it is generally accepted that a tissue eosinophil count >10 per high power field is indicative of this diagnosis. There is growing evidence demonstrating correlations between eCRS severity and inflammatory markers, including blood eosinophilia, eosinophil to total white cell count ratio, and low erythrocyte sedimentation rate [57]. Because of its utility as a marker of lower airway inflammation, FeNO is now being studied as a predictor of eCRS disease severity [72]. Close correlation was found between FeNO and Lund–Mackay CT scores in eCRS patients, whereas FeNO and blood eosinophil count were noted to decrease following functional endoscopic sinus surgery [72]. Of note, there is no increased association between eCRS and serum-specific IgE as measured during immunoCAP testing relative to non-eCRS patients [73]. As mentioned earlier, recent studies have suggested the greater importance of elevated IgE locally, in the pathogenesis of CRS [37,43].

Finally, complimentary testing to identify upper airway disease is essential for asthmatic patients at risk of more severe disease due to the high comorbidity of CRSwNP with asthma [63]. The diagnosis of CRS based on the aforementioned symptom criteria is highly sensitive but inadequately specific [56]. Thus, evaluation must also consist of objective assessment including nasal endoscopy to identify purulence, polyps, or edema or radiographic imaging findings to evaluate for inflammation or mucosal changes within the sinuses [56].

## 9. Asthma Management

At present, the management of asthma is centered on two concepts: optimizing symptom control and improving objective measures of disease severity. Asthma control consists of the minimization of both daytime and nighttime symptoms, maintenance of a normal level of activity, limiting rescue bronchodilator use, and minimizing untoward events such as severe asthma exacerbations. Severity is the intrinsic intensity of the underlying disease process, which precipitates initial treatment choices and future adjustments. The goal of asthma management is to maintain good symptom control over time with the lowest dose of medications necessary and with the fewest side effects [63]. To achieve this end, a stepwise control-based approach is used, in which pharmacologic treatment is adjusted based on a continuous cycle of assessment, treatment, and review of response [63,74].

Medications for asthma are broadly categorized as long-term control medications, used to achieve and maintain control of persistent asthma, or relieving medications used to treat acute symptoms and exacerbations. Initial controlling medication(s) are selected based on disease severity, which is classified as intermittent, mild persistent, moderate persistent, and severe persistent. For intermittent asthma, short-acting beta-2 agonists (SABA), while not a controlling medication, can be used on an as-needed basis, and can generally control these infrequent symptoms. For mild persistent asthma, low-dose inhaled corticosteroids (ICS) are the cornerstone of treatment. ICS have been shown to reduce asthma symptoms, increase lung function, improve quality of life, and reduce the risk of exacerbations, asthma-related hospitalizations, and death [75,76,77,78]. Leukotriene receptor antagonists, while less effective than ICS, can be used in the setting of intolerable adverse effects from ICS, aspirin sensitivity, or with concomitant allergic rhinitis [79,80,81]. With increasing severity, higher doses of ICS are used along with ICS and long-acting beta-2 agonist (LABA) combination medications. Lastly, patients with persistent symptoms or exacerbations despite optimized therapeutic regimens are considered for add-on treatments that include: long acting muscarinic antagonists (tiotropium), low-dose oral corticosteroids, bronchothermoplasty, and biologic therapy [63]. In particular, the use of several novel biologic agents has resulted in improved lung function, reduced the frequency of severe exacerbations, curtailed use OCS, and improved quality of life in refractory patients with T2-high inflammatory patterns [82,83,84,85,86,87,88,89]. Anti-IgE (Omalizumab) therapy has shown benefit for those with severe allergic asthma [84]. Anti-IL-5 (Mepolizumab, Reslizumab), anti-IL-5 receptor (Benralizumab), and anti-IL-4 receptor (Dupilumab) therapy can be used for treatment of uncontrolled, severe eosinophilic asthma [88,90]. To date, biologic therapies for asthma treatment remain an area of active development. Ongoing investigations seek to determine whether these agents have efficacy in the treatment of patients with CRS with nasal polyps.

As previously noted, the presence of severe asthma should warrant careful consideration of other comorbid conditions [91]. Upper airway comorbid diseases are prevalent in severe asthma, which include: rhinosinusitis, obstructive sleep apnea, vocal cord dysfunction, and gastroesophageal reflux disease. The conditions contribute to worsen asthma control, patient quality of life, and complicate diagnostic assessment and treatment of asthmatic patients [92]. Of particular interest is the treatment of upper airway disease such as allergic rhinitis and chronic rhinosinusitis. Of note, further discussion regarding the impact of management of chronic rhinosinusitis is to follow in a later section. Allergic Rhinitis and its Impact on Asthma (ARIA) evidence-based guidelines recommend the routine use of intranasal corticosteroids (INCS) in patients with allergic asthma [93]. Treatment of rhinitis with INCS has been found to improve asthma outcomes, but only in patients with intermittent disease not receiving ICS [94]. Allergen-specific immunotherapy has been shown to improve symptom severity, reduce the use of medications, and to reduce BHR in mild but not severe asthma [64]. For adult patients with allergic rhinitis and sensitization to house dust mite, persistent asthma requiring ICS (FEV1 > 70%) sublingual allergen immunotherapy (SLIT) can be considered, as it showed benefit in decreasing mild to moderate asthma exacerbations [95].

## 10. Chronic Rhinosinusitis Management

Given the robust association between asthma and CRS, the question is raised whether treatment of one disease impacts control of the other disease. Medical therapies including saline irrigation, intranasal and systemic glucocorticoids, antibiotics, and anti-leukotriene agents are used in the treatment of CRS with and without nasal polyposis. A placebo-controlled trial of nasal mometasone in patients with CRS and poorly-controlled asthma showed benefit in asthma symptoms with no benefit for asthma outcomes. This suggests that treatment should be targeted to the symptoms of rhinosinusitis, which may contribute to respiratory symptoms, rather than measures that improve asthma control [96].

Biologic agents that have also been studied for the treatment of CRS with NP include omalizumab, mepolizumab, and dupilumab. In a randomized, double-blind, placebo-controlled trial, omalizumab demonstrated improvements in polyp size, Lund–Mackay score, nasal congestion, anterior rhinorrhea, anosmia, wheezing, and dyspnea [38,97]. In another multicenter, double-blind, randomized control trial (RCT) of mepolizumab, Bachert et al. found a reduction in the need for surgery, and improved visual analog scale (VAS) scores of nasal polyposis, endoscopic polyp scores, and self-reported quality of life (SNOT-22) [38,98]. In a similar study design, Bachert also found the use of dupilumab effective in reducing the polyp burden, Lund–Mackay scores, and peak inspiratory flow of CRSwNP patients [99]. Other novel targets currently under investigation include GATA-3, a transcription factor that is active in the production of IL-4, IL-5, and IL-13 in Th2 cells, and the Singlec-8 receptor, which has been shown to induce apoptosis of eosinophils and inhibition of mast cells [37,98,100,101]. Despite this, none of these drugs are yet approved for the treatment of CRS in the United States. In addition, to date, no studies have been performed to examine whether these agents may be used in combination with each other.

With respect to the AERD endotype of CRSwNP, directed therapies have traditionally included the use of leukotriene modifying agents (LTR antagonists and 5-lipoxygenase inhibitors). Aspirin-desensitization therapy and adherence to a low-salicylate diet have also proven to be useful adjunctive measures and are associated with improvement of CysLT and IL-4 levels [102,103]. Despite this, there still remains a subset of patients with AERD in whom these measures are not efficacious, suggesting that there are subendotypes of this disease that warrant further immuno-pathophysiologic characterization [102].

When CRS is recalcitrant to medical therapy, endoscopic sinus surgery (ESS) is considered. Recently, Schlosser et al. demonstrated in a multi-institutional prospective study that patients with pre-existing asthma and CRS experience improved asthma-specific quality of life (QOL) and asthma control after ESS. Chen et al. previously examined asthma control test (ACT) outcomes after ESS and failed to show improvement in mean postoperative ACT scores [104]. Interestingly, the study cohorts differed dramatically in the number of patients that had poorly-controlled asthma, 51% versus 11% respectively. This suggests the benefit of surgery is most evident in patients with poorly-controlled asthma in the pre-operative setting. This is further supported by recent meta-analyses, which found that ESS in patients with concomitant asthma improves clinical asthma outcome measures and objective and subjective nasal outcomes, but fails to show a benefit in pulmonary function testing [105,106]. Recent studies have also reported that early ESS for symptomatic CRS may decrease the development of asthma [107,108,109]. The otolaryngologist can therefore be of significant help in the difficult-to-treat asthma patient by means of diagnosis and treatment of concordant CRS.

## 11. Conclusions

Asthma and CRS constitute a group of disorders with varying severity, phenotypic expression, and pathogenesis that are often comorbid and difficult to treat. This review of the literature on their concordance suggests that the optimization of management of each may have a substantial impact on the clinical control of the others. The treatment strategies for concomitant disease are still being elucidated, but novel biologic agents show tremendous promise in preliminary studies. Currently, clinical efforts are best directed at the accurate diagnosis of each condition (including endotyping), symptom control, and disease maintenance through best-practice guidelines for each disease process.

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
