# Peer review of "Asthma and Chronic Rhinosinusitis: Diagnosis and Medical Management"

_medsci, 2019, doi:10.3390/medsci7040053_

Round 1
Reviewer 1 Report
The authors have described clinical management and diagnosis clearly for asthma and rhinitis. In particular, authors have defined well asthma as a a heterogenous condition, with two major patient groups being non-allergic asthma and allergic asthma.
While the authors have described IgE as the major mediator in allergic asthma, little is mentioned with regards to the development of IgE in allergic asthma. Although it is understood that the provenance of IgE remains under debate, it would help the readers to gain more knowledge intros field with a couple of references (Line 84):
1) Dullaers et al. The Who, where and when of IgE in allergic airway disease (DOI: https://doi.org/10.1016/j.jaci.2011.10.029)
2) Gould and Wu. IgE repertoire and immunological memory: compartmental regulation and antibody function (doi: 10.1093/intimm/dxy048).
Similarly, diagnosis of allergic asthma and management sometimes rely on detection of allergen specific IgE using ImmunoCAP ISAC platform. The authors should add appropriate information accordingly (http://www.questdiagnostics.com/testcenter/testguide.action%3Fdc%3DTS_ImmunoCapIgE).
Author Response
Point 1: An additional discussion was included (lines 89-95) regarding the theories on the development of IgE mediated disease in allergic asthma.
Point 2: A reference was made to the use of ImmunoCap in confirming the diagnosis of allergic asthma (lines 175-176).
Reviewer 2 Report
Although i have no major qualms with the accuracy of the content, the article underwhelms and does not really provide anything very new. It also fails to deliver on the title as there is no discussion about the diagnosis of CRS and furthermore the focus is towards asthma throughout. There is also some deviation into allergic rhinitis. Some recent relevant studies are missing from the citations.
Author Response
Additional discussion was included about the clinical features, diagnostic criteria, and management of sinusitis, including a greater focus on eosinophilic chronic rhinosinusitis. This discussion included a review of recent studies that have looked at the inflammatory markers associated with this disease.
Reviewer 3 Report
Minor Comments:
- page 1, lines 17-19: the bibliography on "united airway disease" should be updated with recent papers such as Front Pediatr. 2017 Mar 3;5:44.
- In the Introduction, the authors should briefly mention anatomic similarities between upper and lower airways (please refer and cite Int J Immunopathol Pharmacol. 2014 Oct-Dec;27(4):499-508)
- page 4, lines 158-160: the authors should better specify that, in asthma, FeNO is also very useful to verify adherence to therapy and to predict upcoming asthma exacerbations; please refer to Int J Immunopathol Pharmacol. 2011 Oct;24(4 Suppl):29-32
- page 5, lines 201-202: the authors should better emphasize the impact of upper airway pathology on the control of asthma, with particular reference of severe asthma; please refer and cite Expert Rev Respir Med. 2017 Nov;11(11):855-865.
Author Response
Comment 1: The bibliography was updated with this reference.
Comment 2: The relevant citation was changed to reflect this.
Comment 3: The relevant citation was updated to reflect this change.
Comment 4: A comment was made to emphasize the impact of upper airway pathology on the control of asthma, with particular emphasis on CRS, OSA, and GERD (lines 231-234).
Reviewer 4 Report
A re-organization is necessary for me to approve
Author Response
The format of this paper was reorganized by separating the clinical features, diagnosis, and management sections into individual sections for asthma and CRS. This will help the reader to more easily differentiate between these disease processes.
Round 2
Reviewer 2 Report
Unfortunately the revised version does not really alter my view on the usefulness of this review. I'm not sure whether it targets a respiratory or ENT audience but feels as though it falls short on either front. The section labelled CRS diagnosis and assessment ventures into a small subplot about a few biomarkers but does not discuss the current diagnostic criteria and work up in an ENT setting.
Author Response
Thank you for your review. I agree with your reading of the previous versions of this paper. This report is intended to examine the current paradigms behind the evaluation and treatment of these separate but related disease processes. It fell short in comprehensively evaluating the current state of the literature in the clinical features, pathophysiology, and diagnosis of CRS and its many subtypes.
In this revision, I have included a more comprehensive discussion of the concept of endotyping in order to demonstrate correlations with current knowledge of the pathogenesis of asthma and other reactive airways diseases. In doing so, I included a review of the cluster analyses that have been performed to help differentiate between the various endotypes. I separated the clinical features section into 2 sections: asthma and CRS. Within the CRS section, a more in-depth analysis of the research approaches toward endotyping CRSwNP is provided. In the CRS diagnosis and assessment section, a more comprehensive review of classical diagnostic strategies is provided, prior to identifying other novel strategies that are currently being evaluated.
It is my hope that you will look more favorably upon this version of the article.
Reviewer 4 Report
The requests for revision have been made
Author Response
Thank you.